# Addressing the ADME Challenges of Compound Loss in a PDMS-Based Gut-on-Chip Microphysiological System

**DOI:** 10.3390/pharmaceutics16030296

**Published:** 2024-02-20

**Authors:** Patrick Carius, Ferdinand Anton Weinelt, Chris Cantow, Markus Holstein, Aaron M. Teitelbaum, Yunhai Cui

**Affiliations:** Department Drug Discovery Sciences, Boehringer Ingelheim Pharma GmbH & Co. KG, 88400 Biberach, Germany; patrick.carius@boehringer-ingelheim.com (P.C.); ferdinand_anton.weinelt@boehringer-ingelheim.com (F.A.W.); chris.cantow@boehringer-ingelheim.com (C.C.); markus.holstein@boehringer-ingelheim.com (M.H.); aaron.teitelbaum@boehringer-ingelheim.com (A.M.T.)

**Keywords:** microphysiological system (MPS), gut-on-chip, polydimethylsiloxane (PDMS), absorption, distribution, metabolism, and excretion (ADME)

## Abstract

Microphysiological systems (MPSs) are promising in vitro technologies for physiologically relevant predictions of the human absorption, distribution, metabolism, and excretion (ADME) properties of drug candidates. However, polydimethylsiloxane (PDMS), a common material used in MPSs, can both adsorb and absorb small molecules, thereby compromising experimental results. This study aimed to evaluate the feasibility of using the PDMS-based Emulate gut-on-chip to determine the first-pass intestinal drug clearance. In cell-free PDMS organ-chips, we assessed the loss of 17 drugs, among which testosterone was selected as a model compound for further study based on its substantial ad- and absorptions to organ chips and its extensive first-pass intestinal metabolism with well-characterized metabolites. A gut-on-chip model consisting of epithelial Caco-2 cells and primary human umbilical vein endothelial cells (HUVECs) was established. The barrier integrity of the model was tested with reference compounds and inhibition of drug efflux. Concentration–time profiles of testosterone were measured in cell-free organ chips and in gut-on-chip models. A method to deduce the metabolic clearance was provided. Our results demonstrate that metabolic clearance can be determined with PDMS-based MPSs despite substantial compound loss to the chip. Overall, this study offers a practical protocol to experimentally assess ADME properties in PDMS-based MPSs.

## 1. Introduction

Microphysiological systems (MPSs) represent biomimetic in vitro devices that allow for complex human cell and tissue models to be cultured under dynamic biophysical stimuli, such as peristalsis-like stretch and continuous perfusion [1]. Isolated single- or interconnected multi-organ chips are seen as promising in vitro technologies that may offer more physiological and human-relevant predictions of absorption, distribution, metabolism, and excretion (ADME)-related parameters of drug candidates, which, in turn, could lead to better pharmacokinetic (PK) and PK–pharmacodynamic (PK/PD) predictions [2,3] and a reduction in the number of animal studies in drug discovery. Although there has been impressive progress made in the field over recent years, well-characterized and validated MPSs for the prediction of the ADME properties of drug candidates are still not available [4]. In this regard, it is essential to define a specific context of use for the evaluation of an MPS, since the success of the intended experiments is determined by the interaction of two factors that ideally would supplement each other: the selection of a specific MPS and the integration of appropriate cellular models [5]. This becomes particularly obvious in the case of MPSs that are produced from polydimethylsiloxane (PDMS), which is a biocompatible, gas permeable, and flexible polymer [6]. PDMS, however, also strongly ab- and adsorbs a wide range of small molecules, including drugs [7,8,9,10,11] and fluorescent probes, such as fluorescein isothiocyanate (FITC) [12] or Nile red [13]. Within the context of in vitro ADME experiments, compound loss must be considered because it could lead to erroneous determinations of metabolic clearance in PDMS-based MPSs. Long-term compound loss of nicotine and cisplatin was, thus, considered in a report of a multi-organ interconnected MPS by experimentally measuring substance concentrations in the outlets of cell-free organ-chips and then integrating these results into a PK/PD model [14]. Grant et al. also described the combination of an experiment with a simulation-based approach to account for compound loss over the course of several days within a two-channel single-organ MPS, which allowed for the simulation of spatial and temporal amodiaquine concentrations [12]. These model-based approaches were used to account for compound loss during multiday incubation periods in PDMS-based MPSs, but they might be impracticable for experiments performed during shorter periods of time. During routine ADME profiling of drug candidates, predictions of drug absorption in the human intestine are often based on bidirectional permeability experiments normally lasting only a couple of hours. In many cases, such experiments are performed using Caco-2 cells cultured on permeable growth supports under static culture conditions, which is still regarded as a regulatory-accepted industry-wide gold standard [15]. However, this cellular model has also been subject to criticism, since it lacks the physiological expression of some drug transporters but, more importantly, also drug-metabolizing enzymes (DMEs), including cytochrome P450 3A4 (CYP3A4) as one of the most abundant DMEs in the human intestine and liver [16,17]. Although the integration of Caco-2 cells in a gut-on-chip MPS has been shown to increase CYP3A4 activity [18], these results were recently challenged in a comparison between Caco-2 and human primary duodenal organoids both cultured within the same MPS [19]. The simultaneous physiologically relevant activity of DMEs and drug transporters within a single MPS, which would be lined with a complex human-relevant cellular model, could allow for the combined measurement of barrier function, including drug transporter activities, as well as metabolism, within a single assay.

The purpose of this study was to test the feasibility of a PDMS-based MPS for the determination of first-pass intestinal metabolic clearance of drugs with substantial ad- and absorptions to PDMS. We utilized the Emulate gut-on-chip system (Figure 1), encompassing Caco-2 cells (epithelial channel) and primary human umbilical vein endothelial cells (HUVECs) (endothelial channel,) as well characterized cellular models. The barrier integrity of this model was probed with reference compounds and inhibition of ATP-binding cassette subfamily B member 1 (also known as multidrug resistance 1 p-glycoprotein (MDR1 P-gp); hereinafter abbreviated as P-gp).

Testosterone was selected as a tool compound from a set of 17 drugs with different physicochemical properties to determine metabolism and/or clearance parameters within the gut-on-chip. This was only made possible through the identification of a method in which ab- and adsorptions to PDMS were experimentally corrected for a compound-specific and chip-specific clearance, which was derived from time-resolved concentration measurements in cell-free organ-chips.

## 2. Materials and Methods

### 2.1. Materials

The marketed drugs used for the experiments in this manuscript were sourced from commercial vendors (Merck Millipore, Darmstadt, Germany). Apafant and BI1234 were provided by the internal compound management of Boehringer Ingelheim, Biberach, Germany. 

### 2.2. General Cell Culture

Caco-2 cells (clone HTB-37) were obtained from the Leibniz Institute DSMZ-German Collection of Microorganisms Cell Cultures (Braunschweig, Germany), and primary human umbilical vein endothelial cells (HUVECs) from pooled donors were obtained from Lonza (Basel, Switzerland). Caco-2 cells were cultured in a T75 cm^2^ cell culture flask in 12 mL Dulbecco’s Modified Eagle Medium (DMEM) containing 20% fetal calf serum (FCS), 1% nonessential amino acids (NEAAs), 2 mM glutamine, and 100 U/mL penicillin, as well as 100 µg/mL streptomycin (Caco-2 culture medium), in a humidified incubator at 37 °C and 95% CO_2_. Subculturing was conducted after the cell layer reached ~80% confluency, usually every seven days. For subculturing, Caco-2 cells were washed with 10 mL PBS without magnesium or calcium before they were treated with 5 mL Accutase^®^ (Merck, Darmstadt, Germany) for a maximum of 15 min or until all cells were detached. The enzymatic reaction was stopped with 8 mL Caco-2 culture medium, and the cells were centrifuged at 300 rcf for 5 min. Caco-2 cells were counted and 0.25 × 10^6^ viable cells were seeded in a new T75 cm^2^ cell culture flask in 12 mL Caco-2 culture medium. All solutions, except for Accutase (room temperature), were warmed to 37 °C before use. Caco-2 cells were not used for more than 20 passages. HUVECs were handled similarly, with a few exceptions. The HUVECs were fed with 15 mL endothelial cell growth medium, with all supplements from Promocell (Heidelberg, Germany) including 2% FCS (HUVEC culture medium). In addition, the HUVECs were detached with Accutase solution for 5 min, and cells were only used from passages 2 to 6.

### 2.3. Organ-Chip Experiments

#### 2.3.1. Organ-on-Chip System

S-1^®^ Organ-Chips from Emulate Inc. (Boston, MA, USA) comprise 2 parallel channels (top and bottom) for cell culture, which are superimposed and separated via a porous membrane, all made of polydimethylsiloxane (PDMS). The 50 µm thick membrane contains pores of a diameter of 7 µm, which are spaced 40 µm apart from each other. In addition, 2 more channels, in which a vacuum can be applied to laterally stretch the cell culture channels, flank the cell culture channels horizontally. The top channel has a size of 1000 × 1000 µm (width × height), a culture area of 28 mm^2^, and a volume of 28 µL. The bottom channel has a size of 1000 × 200 µm (width × height), a culture area of 24.5 mm^2^, and a volume of 5.6 µL. Both channels share an area of 17.1 mm^2^ of the porous membrane as a coculture area. To conduct experiments, each organ-chip is inserted into a cell culture container called Pod^®^, which connects 4 physically separated fluid reservoirs with the in- and outlets of the top and bottom channels. A maximum of 12 Pods^®^ (including one chip each) can be inserted into the cell culture module Zoë, which resides in a humidified incubator at 37 °C and 95% CO_2_. The Zoë module controls, as well as regulates, the vacuum and pressured gas flows generated by the Orb hub module, which also powers the Zoë module.

#### 2.3.2. Caco-2/HUVEC Gut-on-Chip Standard Culture

The Caco-2/HUVEC gut-on-chip was prepared according to the Caco-2/HUVEC gut-on-chip protocol provided by Emulate Inc. Briefly, on day −1 of the protocol, the PDMS membrane was activated using Emulate reagent (ER) 1 dissolved in ER 2 solution at a concentration of 0.5 mg/mL. The top channels were filled with 50 µL and the bottom channels with 20 µL of ER 1 solution and then treated under UV light for 10 min. Next, activated ER 1 solution was aspirated from both channels and exchanged with fresh ER 1 solution after which the chips were again exposed to UV light for 5 min. The two channels were then subsequently washed with ER 2 solution followed by ice-cold PBS. After the PBS was fully aspirated from both channels, the chips were coated with 100 µg/mL Matrigel and 30 µg/mL collagen type I in PBS overnight at 37 °C. On day 0 of the protocol, both channels were washed with HUVEC culture medium before 0.14 × 10^6^ viable HUVECs in 20 µL HUVEC culture medium were seeded in the bottom channel. The chips were inverted and HUVECs were allowed to adhere for at least 2 h to the porous PDMS membrane. Afterwards, the chips were inverted again, and the top channel was filled with enough Caco-2 culture medium to also cover the in- and outlets (~100 µL). Caco-2 cells were seeded in the top channel (0.15 × 10^6^ viable cells in 50 µL Caco-2 culture medium). Caco-2 cells were also allowed to adhere for 2 h and then the channels were filled with the respective media and incubated overnight at 37 °C. On day 1, the inlet reservoirs of the Pods were filled with 3 mL of the respective degassed media, and the chips were connected to the Pods. The Pods were then placed into the Zoë module, and a flow at a rate of 30 µL/h (manufacturer’s instruction) was applied for both channels. From day 3 onwards, the HUVECs were cultured in HUVEC culture medium but with 0.5% FCS. On day 3, 10% lateral, peristalsis-like stretch was activated with a frequency of 0.15 Hz (manufacturer’s instruction). On day 8, the FCS concentration in the Caco-2 culture medium was reduced from 20% to 10%. Medium reservoirs were replenished with fresh medium 2–3 times per week. Bidirectional permeability assays were performed on day 10, and time-resolved testosterone concentration experiments were performed on day 13 using the same set of chips for both experiments. 

### 2.4. Compound Loss Experiments in Cell-Free Organ-Chips

Compound loss experiments in cell-free organ-chips were performed with used chips after removal of the cells and washing-out of previously perfused compounds (Section 4). The chips, after their use as described in Section 2.3 and Section 2.5, were washed with 50 µL organic solvent (45% acetonitrile/45% methanol/10% double-distilled H_2_O, *v*/*v*/*v*) and stored dry until re-use. For the compound loss experiments in cell-free organ-chips, the chips were washed with PBS without calcium and magnesium three times, inserted into the Pods^®^, and then placed in the Zoë module following the same procedure as described in the Caco-2/HUVEC gut-on-chip protocol provided by the supplier. Degassed transport buffer (128.13 mM NaCl, 5.36 mM KCl, 1 mM MgSO_4_, 1.8 mM CaCl_2_, 4.17 mM NaHCO_3_, 1.19 mM Na_2_HPO_4_, 0.41 mM NaH_2_PO_4_, 15 mM 2-[4-(2-hydroxyethyl) piperazin1-yl] ethane sulfonic acid (HEPES), 20 mM glucose, and 0.25% bovine serum albumin (BSA), pH 7.4) (50 µL/h) through both channels for at least 24 h before the compound loss experiments were conducted. Combinations of the drugs displayed in Table 1 were selected for different molecular weights and perfused as cocktails of a maximum of 4 drugs per cocktail at a concentration of 10 µM of individual compounds in transport buffer through both channels simultaneously. Dosing was initiated via a drug introduction step with a flow rate of 600 µL/h for 10 min (manufacturer’s instruction), after which outlets were aspirated. The compound concentrations in the outlets were measured via liquid chromatography-tandem mass spectrometry (LC-MS/MS, Section 2.8) after a period of one hour, with a flow rate of 200 µL/h (manufacturer’s instruction), and relatively compared to the inlet concentrations, and the difference was reported as the percent drug absorbed. The experiments were performed in duplicate.

### 2.5. Testosterone Compound Loss Profiles

To determine the chip-specific clearance of testosterone, time-resolved compound loss profiles in intervals of 15 min were generated in cell-free organ-chips and the Caco-2/HUVEC gut-on-chip. The cell-free organ-chips were previously utilized in experiments with cells and were prepared in the same way as described for the compound loss studies above. The Caco-2/HUVEC gut-on-chip was cultivated under the conditions described for standard cell culture until day 13. The in- and outlets of the Caco-2/HUVEC gut-on-chip were washed with 1 mL prewarmed and degassed transport buffer prior to dosing testosterone. Testosterone (10 µM) in transport buffer was initially perfused through the top channel via an introduction step (flow rate of 600 µL/h for 10 min), while the bottom channel was flushed with transport buffer at the same flow rate. An introduction step with a high flow rate of 600 µL/h is recommended by the manufacturer to “saturate” the ad- and absorptions of compounds by the PDMS material. The necessity of this step will be discussed in Section 3 and Section 4. After the introduction step, both outlets were aspirated. For a total duration of 135 min (flow rate: 200 µL/h), the top in- and outlets and the bottom outlet, as well as the complete volume of each channel from a single chip per timepoint were sampled every 15 min. During the sampling, the flow was stopped for a few seconds, a Pod^®^, which included a chip, was removed from the Zoë, a single chip removed from the Pod^®^, and samples were taken. The samples from the bottom outlet were used directly for bioanalytical quantification. The samples from the top inlet and top outlet were sampled from the respective reservoirs and diluted 1:50 (*v*/*v*) in transport buffer. The chips were subsequently dismantled from the Pods, and samples from the top channel and bottom channel were taken with a pipette via the outlets (Figure 1A). The samples from the top channel were diluted 1:50 (*v*/*v*) and the samples from the bottom channel 1:2 (*v*/*v*) in transport buffer. All samples were then spiked with the internal standard and used for the LC-MS/MS analysis, as described in Section 2.8.

### 2.6. Bidirectional Permeability Assays

Either the top (A–B direction) or bottom (B–A direction) channel served as the donor compartments and the respective other channels (A–B: bottom; B–A: top) as receiver compartments. Apafant, as a known substrate for P-gp, and BI1234, a proprietary low-permeability control (P_app_ ≈ 3 × 10^−^^7^ cm/s; no efflux), were chosen as reference compounds to probe the P-gp activities and barrier function of the Caco-2/HUVEC gut-on-chip [17,20,21]. Following an Emulate protocol, dosing of a cocktail of both compounds at 10 µM in transporter buffer was initiated via a drug introduction step (flow rate of 600 µL/h for 10 min), after which the outlets were aspirated. The outlet concentrations were determined after one hour (flow rate: 200 µL/h). To achieve P-gp inhibition in the respective experiments, the P-gp inhibitor elacridar [22] was pre-incubated for 75 min at a flow rate of 200 µL/h. The Apafant and BI1234 cocktail was then dosed together with elacridar at 200 µL/h for one hour. If not indicated otherwise, for the bidirectional permeability experiments, 3 chips were used per group for the following 4 groups: A–B, B–A, A–B + 10 µM elacridar, and B–A + 10 µM elacridar. The samples from the receiver outlets were used for the bioanalysis without dilution. The samples from the donor inlets, as well as outlets, were diluted 1:50 (*v*/*v*) in transport buffer. All samples were spiked with the internal standard and used for the LC-MS/MS analysis, as described in Section 2.8. The permeability in Caco-2 cells grown on Transwell^®^ (Corning, Kaiserslautern, Germany) inserts was measured as described previously [17,20,21].

The apparent permeability coefficient (P_app_) was either calculated for molecular transport in the A–B direction (donor: top channel, receiver: bottom channel) or B–A direction (donor: top channel, receiver: bottom channel) using Equation (1) [23]:(1)$$P_{app}=-\frac{Q_{R}\times Q_{D}}{SA\times \left(Q_{R}+Q_{D}\right)}\times ln\left[\frac{C_{R,0}\times \left(Q_{R}+Q_{D}\right)}{\left(Q_{R}\times C_{R,0}+Q_{D}\times C_{D,0}\right)}\right]$$
where Q_R_ is the flow rate in the receiving channel, Q_D_ is the flow rate in the dosing channel (both = 200 µL/h), SA is the coculture surface area of each chip (0.171 cm^2^), C_R,0_ and C_D,0_ are the recovered receiver or donor channel outlet concentrations, respectively.

The efflux ratio was calculated as follows (Equation (2)): (2)$$Efflux\;ratio=\frac{P_{app,B–A}}{P_{app,A–B}}$$
where P_app,A–B_ is the permeability coefficient for a compound from the top channel to the bottom channel, and P_app,B–A_ is the permeability coefficient in the opposite direction. 

### 2.7. Metabolite Identification

The metabolites formed in the Caco-2/HUVEC gut-on-chip were analyzed after conducting the same protocol as already described for the testosterone compound loss profiles over time. In brief, after testosterone (10 µM) was perfused through the top channel at a flow rate of 600 µL/h for 10 min, samples were collected after two hours and four hours from the top, as well as bottom, outlets of the cell-free organ-chips or the Caco-2/HUVEC gut-on-chip (200 µL/h). The samples were mixed with the same amount of 0.1% formic acid in acetonitrile, subsequently evaporated, and then resuspended in water containing 25% methanol and 0.1% formic acid. The analysis was performed on an LC-MS system containing a Vanquish UPLC (ThermoFisher Scientific, San Jose, CA, USA) coupled to an Orbitrap FusionTribrid high-resolution mass spectrometer (ThermoFisher Scientific). The structural elucidation was based on exact mass measurements in combination with the interpretation of fragment spectra.

### 2.8. Bioanalytical Quantification of Tested Drugs and Metabolites

A quantitative bioanalysis was performed with a fit-for-purpose assay using protein precipitation followed by LC-MS/MS. The equipment involved an HPLC series 1000 or higher from Agilent (Santa Clara, CA, USA) and the mass spectrometers API 6500 or higher from AB Sciex (Toronto, ON, Canada) operating in the multiple reaction monitoring (MRM) mode.

## 3. Results

### 3.1. Assessing Compound Loss in Cell-Free Organ-Chips

PDMS is a porous and hydrophobic material that ad- and absorbs a variety of small molecules. Although combinations of some physicochemical molecular properties (molecular weight < 1 kDa [12]; mean ClogP > 2.7 [8]; TPSA < 50 A^2^ [24]) of small molecules have been discussed to increase the chances of ad- or absorption to PDMS, compound loss within in vitro assays seems to be influenced by additional factors, such as solute/solvent pairings and compound concentrations, as well as residence time [11]. To account for the specific in vitro environment of the Caco-2/HUVEC gut-on-chip, we performed initial perfusion experiments (one hour; flow: 200 µL/h) in cell-free organ-chips to estimate the amount of PDMS compound loss of a set of 17 small molecules (10 µM; perfused simultaneously via the top and bottom inlets), which covered a range of different physicochemical molecular properties (Table 1). The PDMS compound loss during the one-hour period ranged from negligible (indinavir) to complete (oxybutynin), calculated from the difference between the inlet and outlet concentrations of the top and bottom channels. On the basis of its loss (35.1% in the bottom channel and 37.6% in the top channel) after one hour in cell-free organ-chips, testosterone was selected as a model compound to determine the metabolic clearance for two reasons: (1) Ideally, a model compound for the discrimination between chip-intrinsic clearance via ad- and absorptions and metabolic clearance in cellular models on chips should exhibit moderate compound loss in cell-free organ-chips. A much higher compound loss to the chip material would make the assessment of metabolic clearance impossible. (2) Since we were using Caco-2 cells for the gut-on-chip model, compounds that are mainly metabolized by enzymes absent in the Caco-2 cells (e.g., CYP3A4) are not suitable. Testosterone is known to be biotransformed into well-characterized metabolites by enzymes expressed in the human intestine and in Caco-2 cells [25,26,27,28].

Interestingly, the two P-gp inhibitors elacridar and zosuquidar, used routinely in our lab, showed strong differences regarding their compound loss to PDMS. With a compound loss close to 100% after 1 h of perfusion, zosuquidar was disqualified for further P-gp inhibition studies (Section 3.2).

### 3.2. Assessing Barrier Integrity and P-gp Activities in Caco-2/HUVEC Gut-on-Chip

Before a first-pass intestinal clearance experiment could be performed, the barrier integrity of the Caco-2/HUVEC gut-on-chip model had to be confirmed. The barrier integrity of the Caco-2/HUVEC gut-on-chip model was probed with the Boehringer Ingelheim internal low-permeability control BI1234 and the P-gp substrate Apafant. With a P_app,A-B_ of 3.7 *×* 10^−^^7^ cm/s and an efflux ratio of 1.9 (Figure 2), BI1234 showed a permeability in Caco-2/HUVEC gut-on-chip comparable to that in Caco-2 on Transwell inserts (3 × 10^−^^7^ cm/s, internal results). The P-gp substrate Apafant showed an efflux of 22.2 in the Caco-2/HUVEC gut-on-chip (Figure 2), which falls in at the higher end of the variability of the efflux measured in Caco-2 cells on Transwell (12.4 ± 7.8, n = 916). The P-gp-mediated efflux of Apafant in the gut-on-chip model was further confirmed in an experiment in the presence of the P-gp inhibitor elacridar (10 µM). Coperfusion with elacridar inhibited the efflux of Apafant completely (Figure 2).

### 3.3. Time-Resolved Testosterone Loss in the Caco-2/HUVEC Gut-on-Chip

A widely applied method to determine the metabolic clearance of a drug in an in vitro system is to measure the time-dependent depletion of the test drug in the system. Any compound loss not related to metabolism, but rather attributed to PDMS ad- and absorptions, would lead to the overestimation of the clearance. To discriminate between ad-/absorption and metabolism-related compound loss in an organ-on-chip system, the time profiles of the testosterone concentrations in different compartments of cell-free organ-chips, as well as the Caco-2/HUVEC gut-on-chip, were measured to understand the time-based changes in the testosterone loss. The chosen time frame represents the duration of a typical in vitro experiment, and, thus, time-resolved measurements of testosterone concentrations were performed for a total of 135 min in 15 min intervals. Chips were perfused with a testosterone solution (10 µM) only via the top inlet. Every 15 min, one chip was removed from the Pod^®^ cradle, and samples were collected from different compartments (top channel + top outlet; bottom channel + bottom outlet) (Figure 3A,B). The testosterone concentrations in the top channel and outlet of the cell-free organ-chips were low after the substance introduction (600 µL/h for 10 min) but then increased over time to a concentration of ~7000 nM. Such a concentration time profile indicates that a plateau may be reached at later time points. The testosterone concentrations in the bottom channel and outlet of cell-free organ-chips showed a slight increase over the duration of the experiment, reaching a final concentration < 1000 nM. In the Caco-2/HUVEC gut-on-chip, the testosterone seemed to already reach a stable concentration at the earliest timepoints in all measured compartments. The maximum concentration, however, remained at <2000 nM even in the top channel and outlet. The differences observed between the cell-free organ chips and the Caco-2/HUVEC gut-on-chip are addressed in detail in the following sections. Sampling from within the channels presents a technical challenge due to the unavoidable risk of cross-contamination between the top and bottom channels, as well as the small volumes available for sampling, particularly from the bottom channel. Thus, the concentrations in the channels were more prone to variability and should be regarded rather qualitatively. They are, anyway, not needed for the calculation of the clearance (Section 3.5).

### 3.4. Testosterone Metabolism in the Caco-2/HUVEC Gut-on-Chip

A plausible explanation for the difference in testosterone concentration between cell-free organ-chips and the Caco-2/HUVEC gut-on-chip, as shown in Figure 3, is the biotransformation of testosterone by the cells. Metabolite identification via mass scan revealed that testosterone was metabolized into 4-androstenedione, testosterone glucuronide, and testosterone sulfate in the Caco-2/HUVEC gut-on-chip (Figure 4). After testosterone solution was perfused for 2 h through the top channel of the Caco-2/HUVEC gut-on-chip, 4-androstenedione was the main metabolite identified in the top and bottom outlets (96.5%), followed by testosterone glucuronide (3.2%). Traces of testosterone-sulfate were also detected (0.3%). The relative amounts of the individual metabolites were calculated with the peak areas of the respective metabolites. In addition, testosterone, 4-androstenedione, and testosterone glucuronide were separately quantified after 2 h in the outlets of the Caco-2/HUVEC gut-on-chip via LC-MS/MS using standards of the respective metabolites and the parent drug (Figure 5).

### 3.5. Calculation of Metabolic Clearance of Testosterone in the Gut-on-Chip

It is important to note that all outlet concentrations shown in Figure 3 represent cumulative concentrations of testosterone, since one individual organ-chip was sampled for each time point and could not be reassembled into the flow system. The outlet reservoirs contained, therefore, perfusates from time zero to time point t. The concentrations in these samples, especially in samples from later time points, are different to the actual concentrations in the outlet at time point t. Given the technical challenges preventing online measurements of the compound concentrations in the outlet, we decided to calculate the compound output for a defined perfusion period based on the results shown in Figure 3. A period of 60 min was chosen because it was recommended by the manufacturer for diverse applications. The concentration of testosterone in the perfusate collected for 60 min, which represents the time point t − 60 to t (minutes), was calculated using the following equation:(3)$$C_{t,\;last}=\frac{{C\;}_{t,\;cum}\times V_{t}-C_{t-60,\;cum}\times V_{t-60}}{Q\times 1h}$$where Q is the flow rate (200 µL/h), and C_t,cum_ represents testosterone concentrations measured at each time point, t, in the cumulative outflow (as shown in Figure 3). C_t,last_ represents the average concentration of testosterone over the last putative collecting period of 60 min. For time points before 60 min, C_t,last_ was identical to C_t,cum_ by definition. V_t_ is the total volume of perfusate collected from time zero to t. V_t_ was calculated with the following equation:(4)$$V_{t}=t÷60\times Q$$

The time profiles of C_t,last_ in the cell-free organ-chips and in the Caco-2/HUVEC gut-on-chip are shown in Figure 6A,B. Using C_t,last_, the compound loss can be calculated in each putative collecting period of 60 min with the following equation:(5)$${Loss}_{t}=\left(C_{inlet}-C_{t,\;last,\;top}-C_{t,\;last,\;bottom}\right)\times Q$$

The time-dependent compound loss for each collecting period in cell-free organ-chips and Caco-2/HUVEC gut-on-chip is shown in Figure 6C. In the Caco-2/HUVEC gut-on-chip, compound loss stayed constant, suggesting that a steady state was reached very early. In contrast, the compound loss in cell-free organ-chips was time-dependent and reached a steady state only at later time points. In cell-free organ chips, the compound loss during the first 60 min of the collecting period was six-fold higher than in the last 60 min of the collecting period (719.1 pmol/h vs. 116.5 pmol/h). The difference between the compound loss in te hCaco-2/HUVEC gut-on-chip and in the cell-free organ-chips was the basis for the calculation of the metabolic clearance associated with the gut model; thus, collecting samples right after the flushing period would strongly underestimate the metabolic clearance. 

The clearances of the testosterone in the cell-free organ-chips and in the Caco-2/HUVEC gut-on-chip were calculated by the following equation:(6)$$Clearance={Loss}_{135}÷C_{inlet}$$
where Loss_135_ represents the compound loss during the putative collecting period between 75 min and 135 min. The testosterone had a clearance of 11.7 µL/h in the cell-free organ-chips (chip-specific clearance) and 172.1 µL/h in the Caco-2/HUVEC gut-on-chip. The metabolic clearance of the testosterone in the gut-on-chip can be calculated by subtracting the chip-specific clearance from the total clearance in the Caco-2/HUVEC gut-on-chip (160.4 µL/h), which was close to the flow rate of 200 µL/h. 

The calculated C_t,last_ also allowed for the calculation of the apparent permeability coefficient (P_app,A-B_) of the testosterone for different collecting periods. As shown in Figure 6D, the calculated P_app_ values (Equation (1)) remained approximately the same during the experiment. This was expected because both the metabolic clearance and the compound loss to chip material could be considered first-order processes, and all first-order processes leading to compound loss were already accounted for by Equation (1) [23,29]. In the Caco-2/HUVEC gut-on-chip, the P_app_ values varied between 8.5 and 21.2 × 10^−^^6^ cm/s, consistent with the P_app,A-B_ of the testosterone in the Caco-2 on Transwell (11.2 × 10^−^^6^ cm/s).

**Figure 6 pharmaceutics-16-00296-f006:**
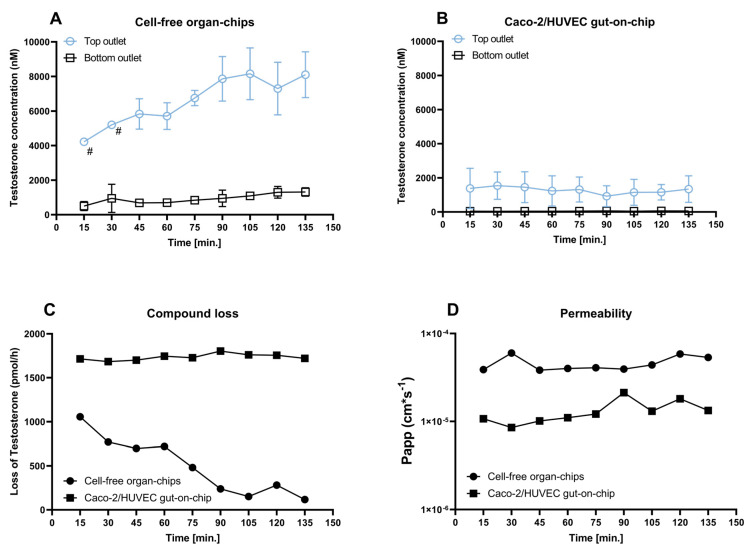
Conversion of cumulative testosterone concentration in the outlet (C_t,cum_) into the concentration in the outflow of a putative 60 min collecting period (C_t,last_) ((**A**) Cell-free organ-chips; (**B**) Caco-2/HUVEC gut-on-chip) and calculation of the compound loss (**C**) and apparent permeability coefficient P_app_ (**D**) in this time period in the organ-chips. (**A**,**B**) For a putative collecting time period of 60 min, the concentration of testosterone in the collected outflow was calculated with Equations (3) and (4). For time points 15, 30, and 45 min, C_t,last_ was identical to C_t,cum_. Data represent the mean ± SD of 3 biological replicates, except the top outlet concentrations at 15 and 30 min (marked with # in (**A**)), which represent the average out of two biological replicates (one replicate at each of these time points was excluded because of technical failure). (**C**) Compound loss in a putative collecting time period of 60 min calculated via Equation (3). For time points 15, 30, and 45 min, the compound loss was extrapolated to a period of 60 min assuming unchanged rates of all processes at this time point. The mean C_t,last_ values from panel (**A**,**B**) were used for the calculation. (**D**) The apparent permeability coefficient (P_app_) for each collecting period was calculated using the equation provided by the manufacturer [30], with C_t,last,top_ as the donor concentration and C_t,last,bottom_ as the receiver concentration.

## 4. Discussion

By introducing peristalsis-like stretch and/or continuous perfusion, MPSs have shown great promise in advancing in vitro technologies for more physiologically relevant predictions of human ADME processes.

The biomechanical forces constantly acting on the human intestinal mucosa in vivo, which, among others, include shear stress, cyclic strain, and cellular deformation, strongly influence cellular biology [30]. Continuous perfusion over multiple days has been shown to enhance the polarization and formation of apical microvilli of Caco-2 cells [31] and primary intestinal epithelial cells derived from duodenal organoids [19], all grown in the Emulate’s gut-on-chip. Combining continuous perfusion with peristalsis-like stretch within the same MPS further led to the formation of intestinal villi-like structures that include characteristic crypt-like invaginations, as shown for Caco-2 cells [18,32] and primary intestinal epithelial cells derived from duodenal organoids [19]. Another recent study further highlighted the importance of basolateral fluid flow facilitated by the MPS as a physiologically relevant clearance mechanism, which is thought to enable the formation of villi-like structures through the removal of morphogen antagonists from the basolateral side [33]. Caco-2 cells that have been subjected to rhythmic deformation in a stretchable in vitro device further showed strain-associated modulation of brush border enzymes [34]. As a biocompatible material, PDMS essentially provides the needed elasticity to allow for peristalsis-like stretch within most stretchable MPSs. However, PDMS also strongly ab- and adsorbs a wide range of small molecules. Our data, which were recorded taking the specific in vitro environment of the Caco-2/HUVEC gut-on-chip into account to determine the substance loss, support recent findings in the same MPS [12]. Although combinations of physicochemical risk factors that are associated with a higher chance of ad- or absorption to PDMS (molecular weight < 1 kDa [12]; mean cLogP > 2.78; TPSA < 50 A^2^ [24]) could provide an early estimate before performing a specific MPS experiment, only an actual measurement of compound loss under the planned experimental conditions provides robust information as similarly discussed in recent literature [6,10,11,12,24]. For example, buspirone with a lower cLogP and larger TPSA compared to quinidine showed higher loss than quinidine. It is interesting to note that some drugs showed different losses in the top and bottom channels. It is currently unclear to us whether experimental variability or differences in the geometry of the channels led to these differences. Since most of the drugs did not show this discrepancy, the difference in the channel geometry does not seem to be a straightforward explanation. However, the surface area of the top channel is more than three-fold larger than that of the bottom channel, and different adsorptions in the top and the bottom channels cannot be excluded. These results strongly suggest that an initial determination of the compound loss in the cell-free organ-chips after an appropriate period of perfusion as a first step when planning ADME-relevant experiments in PDMS-based MPSs as a time efficient step to filter out molecules that show a high risk for compound loss within the chosen setup. Additionally, we want to emphasize that the cell-free organ-chips can be re-used from experiments with cells, after the cells have been lysed and the chips have been prepared as described in Section 2. After a 24 h washout period, we could not detect any residual drugs in the cell-free organ-chips when maximum drug concentrations of 10 µM were used over a 135 min assay window in the previous experiment. We also tested the commercially available compound distribution kit from Emulate, which has the same design as the S1-chip with the difference that the two channels are separated by a PDMS membrane without pores. Our data indicate a negligible difference in the compound loss when comparing results from the compound distribution kit or the re-used cell-free organ-chips. To our surprise, the experiments with the compound distribution kit further showed that the PDMS membrane without pores does not hinder drug permeation between the top and bottom channels (Appendix A).

To study the interplay between the metabolic clearance and chip-intrinsic clearance (PDMS ad- and absorptions) of a compound within a specific MPS, testosterone was selected as a model compound with substantial compound loss (Table 1) and well-characterized metabolic pathways from the set of 17 drugs. As a second step in our proposed protocol, we recommend time-resolved concentration profiles measured in different compartments (top and bottom channels and respective outlets) of cell-free organ-chips (Figure 3B). In the case of testosterone, these showed that the concentration in the top outlet only gradually increased during the first 75 min of the experiment before presumably developing into a plateau. The plateau was more obvious when the testosterone concentration was calculated for every perfusion period of 60 min (Figure 6A). In order to understand the concentration–time profile of the testosterone in the cell-free organ-chips, we needed to dissect the clearance of the testosterone in the organ chips into different routes: testosterone concentration in a perfused MPS initially increased because of the input (C_inlet_ × Q) being higher than the compound loss caused by the various clearance pathways (C_channel_ × (CL_PDMS_ + CL_perm_ + Q)), with CL_PDMS_ representing clearance via PDMS ad- and absorptions and CL_perm_ representing clearance via permeability toward the bottom channel and Q the flow rate. Eventually, the testosterone concentration in the chip would approach a steady state when the compound input and output hold a balance. The time to steady state is determined by the clearance and the volume of distribution (CL/V). For a given compound and a given MPS geometry, the volume of distribution can be considered constant. The only parameter that has a major impact on both the time to steady state and the steady-state concentration in the top channel is the compound clearance in the chip. This could explain why the testosterone concentration in the Caco-2/HUVEC gut-on-chip appeared constant during the whole perfusion period. The additional clearance via metabolic turnover by Caco-2 cells in the gut-on-chip compared to the cell-free chips is so high that a steady state was achieved already after 15 min (first sample time point). Indeed, the total clearance of testosterone in the Caco-2/HUVEC gut-on-chip was 172.1 µL/h, which was close to the perfusion rate of 200 µL/h. This suggests that the clearance of testosterone in the Caco-2/HUVEC gut-on-chip could be limited by the perfusion rate. Consistent with the much higher clearance in the Caco-2/HUVEC gut-on-chip, the steady-state concentration of testosterone in the top outlet was also much lower than in the cell-free organ-chips (Figure 3 and Figure 6). When compared with the cell-free organ-chips, the volume of distribution of the testosterone should be higher in the Caco-2/HUVEC gut-on-chips due to distribution into the cells. Our data show a low cellular distribution of testosterone. Together with the negligible total volume of the cells compared to the volume of the chip material, the volume of distribution of the testosterone in the cell-free organ-chips and in the Caco-2/HUVEC gut-on-chips could be considered identical. This is not true for compounds with extensive cellular distribution. The time to steady state for such compounds in cells containing organ-chips could be much longer. On the basis of these considerations, we conclude that (1) the clearance of a test compound within a PDMS-based MPS could be calculated easily with the outlet concentration under steady-state conditions, and (2) the time to steady state for each test compound could vary because of a different clearance within specific MPS setups and needs to be determined by a concentration–time profile measurement. Since the determination of the metabolic clearance when using other cellular models within the same MPS would include a clearance determination in cell-free organ-chips, determining the optimal starting time point for sample collection would be essential for every other model. 

We must note that the clearance concept described above, especially the parameter CL_PDMS_, is not suitable to discriminate PDMS ad- from absorption. It has been hypothesized that compound loss to PDMS could be described by a fast adsorption process to the channel surface and a rather slow absorption process into the bulk PDMS [12,35,36]. To describe all these processes in a mathematical model, the channel surface adsorption could be described as a distribution process which would be included in the parameter “volume of distribution”; hence, a higher surface area adsorption would increase the volume of distribution. Absorption into the bulk PDMS, however, could be considered a clearance process, since the back diffusion is probably very low due to the huge sink capacity of the bulk PDMS compared to the volume of the perfusion channels. In light of these considerations, one could ask whether a compound introduction step with a higher flow rate (600 µL/h for 10 min), as recommended by the manufacturer, is necessary. The purpose of this step is to saturate the compound adsorption on the channel’s surface. Since we recommend the determination of metabolic clearance under steady-state conditions, an introduction step would not be necessary; it could, nevertheless, shorten the time to reach the steady state. 

These considerations also led to a modification of the experimental protocol for the bidirectional transport experiments with the P-gp inhibitor elacridar. According to the manufacturer, compound loss with first-order kinetics is accounted for by the Equation (1) [23,29]. Indeed, the permeability of testosterone seems to be independent of different compound losses at different time points (Figure 6D). However, the effect of an inhibitor in the organ-on-chip models is expected to depend on its actual concentration in the chip. Elacridar showed compound loss to PDMS to the same extent as testosterone (Table 1). We, thus, suspected a similar time-dependent loss of elacridar in the gut-on-chip. Although the measurement of the permeability of the probe substrates would be independent of the starting time point (Figure 6D), the concentration of the inhibitor within the chip would be lower right after the flushing period than during the steady-state phase (Figure 6C). We, thus, introduced a pre-incubation period (75 min) for elacridar. Given its rather moderate loss to PDMS, high potency (IC90 ~1 µM, Appendix A), and the rather high concentration used in our experiment (10 µM), we did not expect a strong difference with or without the preincubation period for elacridar. However, inhibitors with substantial loss to PDMS (e.g., zosuquidar, see Table 1) would need to be used either at a much higher concentration than in a static model or preincubated accordingly in organ-chips. 

The cultivation of Caco-2 cells under the influence of peristalsis-like stretch in the same MPS was further reported to show a four-fold increase in the paracellular permeability of 20 kDa FITC-labeled dextran when compared to static or perfusion-only controls [31]. Interestingly, in our study, the barrier function of Caco-2 cells grown in the Caco-2/HUVEC gut-on-chip was comparable to historical results obtained by Transwell when using BI1234 (in-house data). In our opinion, these different results could be related to the use of a different Caco-2 clone (BBE) in the aforementioned study compared to the Caco-2 clone HTB-37 that was used in our study. Although their formation was reproducible, we also did not observe as many and dense villi-like structures as reported in other studies (Appendix A).

In accordance with the literature, we identified 4-androstenedione as the main metabolite of testosterone formed by Caco-2 cells, most likely via 17β hydroxysteroid dehydrogenase type 2, which was also the case for ex vivo models of human intestinal mucosa and in vitro models of primary intestinal epithelial cells [26,37,38,39,40,41]. Testosterone glucuronide was identified as the second most abundant testosterone metabolite in our study (Figure 5). Testosterone glucuronide formation has been shown for well-differentiated Caco-2 cells, and the abundance of the metabolite identified by us is approximately two-fold lower than reported for primary enterocytes [26,40]. However, as previously reported elsewhere and confirmed by us in this study (Appendix A), the identified metabolites are also subjected to compound loss [7]. Nevertheless, also the concentrations of the metabolites seem to reach a steady state toward later timepoints of the experiment (Appendix A). Furthermore, we did not detect any CYP3A4-related hydroxylation of testosterone in the Caco-2/HUVEC gut-on-chip, as reported by others [19]. With the concentrations of the parent drug and the two major metabolites in the top and bottom outlets, we were also able to calculate the mass balance for the Caco-2/HUVEC gut-on-chip model. In the last perfusion period (120–135 min), the recovery of testosterone, 4-androstenedione, and testosterone glucuronide was 21%, 27%, and 2%, respectively. Including the chip-intrinsic clearance of testosterone (11.7 µL/h, corresponding to a compound loss of 6%), the total recovery of all drug-related materials was 56%. Not included in this calculation are the chip-intrinsic clearance of both metabolites. The main metabolite, 4-androstenedione, seems to have a much higher compound loss in cell-free organ-chips (Appendix A), it is reasonable to assume that the total recovery should be much higher than 56%. In addition, metabolic steps secondary to 4-androstenedione are possible. Further studies to elucidate these aspects are required. 

In this study, Caco-2 cells together with HUVECs were used as a well-characterized cellular model for setting up the basic protocol for clearance studies. We must emphasize that Caco-2 cells are not an appropriate model for the human intestine due to the lack of major drug-metabolizing enzymes like CYP3A4. The aim of this study was not to validate the Caco-2/HUVEC gut-on-chip model itself, but rather to provide a basic protocol for the determination of metabolic clearance in an MPS made of PDMS. On the basis of its nature, this protocol is suitable for any cellular model (e.g., liver-on-chip) in which the metabolic clearance should be determined. Fully characterized gut-on-chip models with human intestinal organoids are unfortunately not yet available. These models are currently being established in our lab. Further studies with these more physiologically relevant models to verify the basic protocol we are proposing here are warranted.

## 5. Conclusions

In sum, our study showed, for the first time, that the metabolic clearance of testosterone, a compound with substantial loss in PDMS-based MPSs, can be determined in a gut-on-chip model made from PDMS. We propose the following basic protocol for clearance studies in PDMS-based MPSs:-Determination of the compound loss within a defined perfusion period (e.g., one hour) to exclude test compounds like oxybutynin, which are heavily cleared by PDMS;-Determination of the time-resolved concentration profiles of test compounds in cell-free organ chips. Here, the goal is to define a time frame in which the concentration of the test compound reaches a steady state in the outlet;-Measurement of the compound loss in cell-free organ chips and in organ chips containing organ models during this period;-Calculation of the metabolic clearance by subtracting chip-intrinsic clearance from the total clearance.

## Figures and Tables

**Figure 1 pharmaceutics-16-00296-f001:**
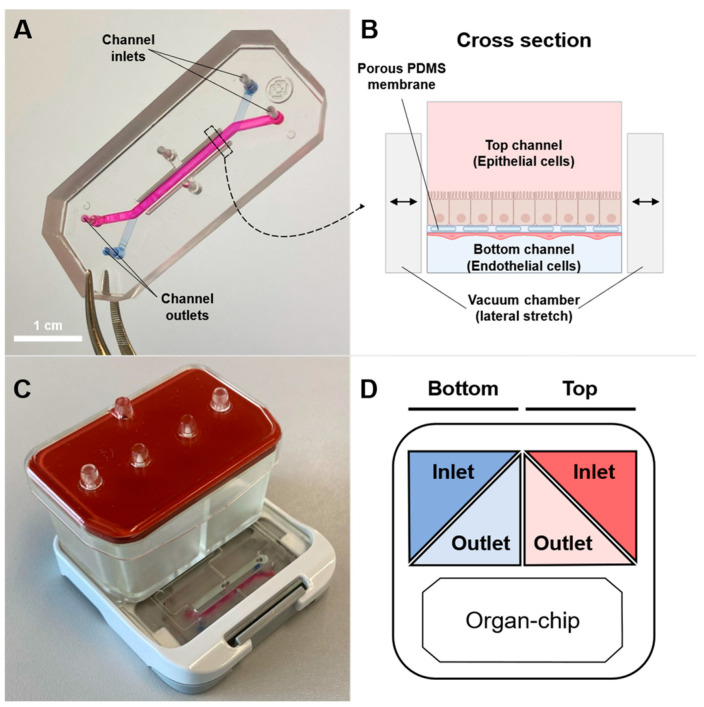
The Emulate gut-on-chip with the top channel highlighted in red and the bottom channel highlighted in blue (**A**). Cross-section of the area marked in (**A**), showing the position of the cells within each channel and the direction of movement of the PDMS membrane (**B**). The organ-chips are inserted into Pods^®^ (**C**), which provide the reservoirs for the respective channel in- and outlets. The positions of the respective reservoirs are schematically shown (**D**). The scheme depicted in (**B**) was created with BioRender.com.

**Figure 2 pharmaceutics-16-00296-f002:**
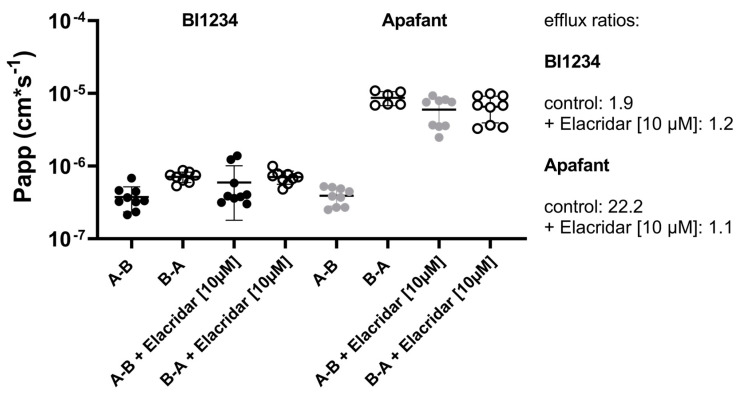
Bidirectional permeability of BI1234 and Apafant in the Caco-2/HUVEC gut-on-chip. Experiments were performed on day 10 of the coculture of Caco-2 cells with HUVECs. In the inhibition experiments, the P-gp inhibitor elacridar (10 µM) was started 75 min before BI1234 and Apafant (both 10 µM) were added as a cocktail. Data represent the mean ± SD of 3 biological replicates, with at least 2 technical replicates per experiment.

**Figure 3 pharmaceutics-16-00296-f003:**
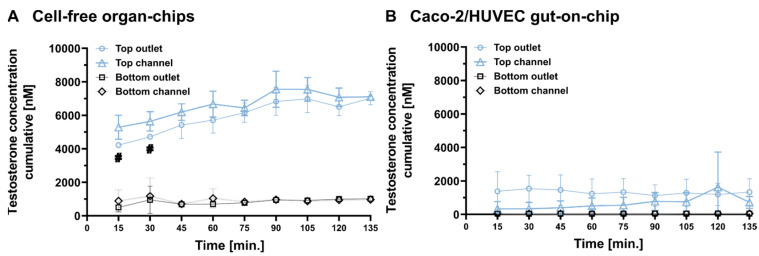
Concentration over time profiles of the testosterone perfused into the top inlet at 200 µL/h at a concentration of 10 µM in cell-free organ-chips (**A**), as well as in the Caco-2/HUVEC gut-on-chip (**B**). Data represent the mean ± SD from three biological replicates, except the top outlet concentrations at 15 and 30 min (marked with #), which represent the average out of two biological replicates (one replicate at each of these time points was excluded due to technical failure). Inlet concentrations of testosterone (11.8 ± 0.8 µM in cell-free organ-chips and 11.0 ± 0.6 µM in Caco-2/HUVEC gut-on-chip) were measured at each time point in both models to exclude compound loss in the inlet reservoirs.

**Figure 4 pharmaceutics-16-00296-f004:**
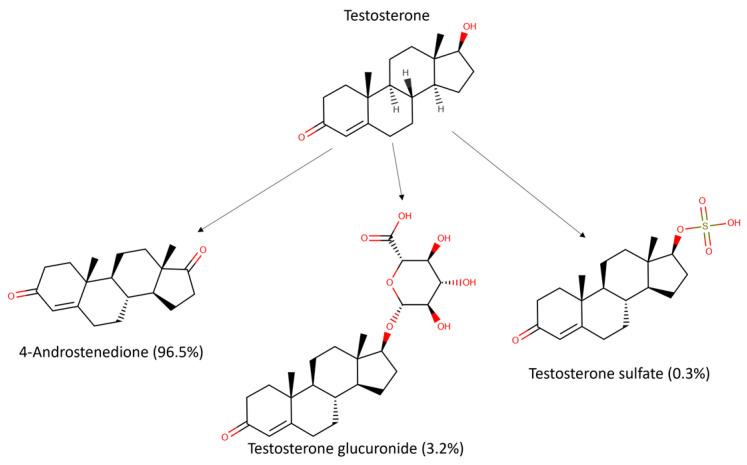
Biotransformation of the testosterone in the Caco-2/HUVEC gut-on-chip. A metabolite scan of the samples from the gut-on-chip perfused with 10 µM testosterone solution at a flow rate of 200 µL/h for 2 h. The values in parentheses represent the fraction of the total amount of all metabolites, calculated with peak areas of the metabolites.

**Figure 5 pharmaceutics-16-00296-f005:**
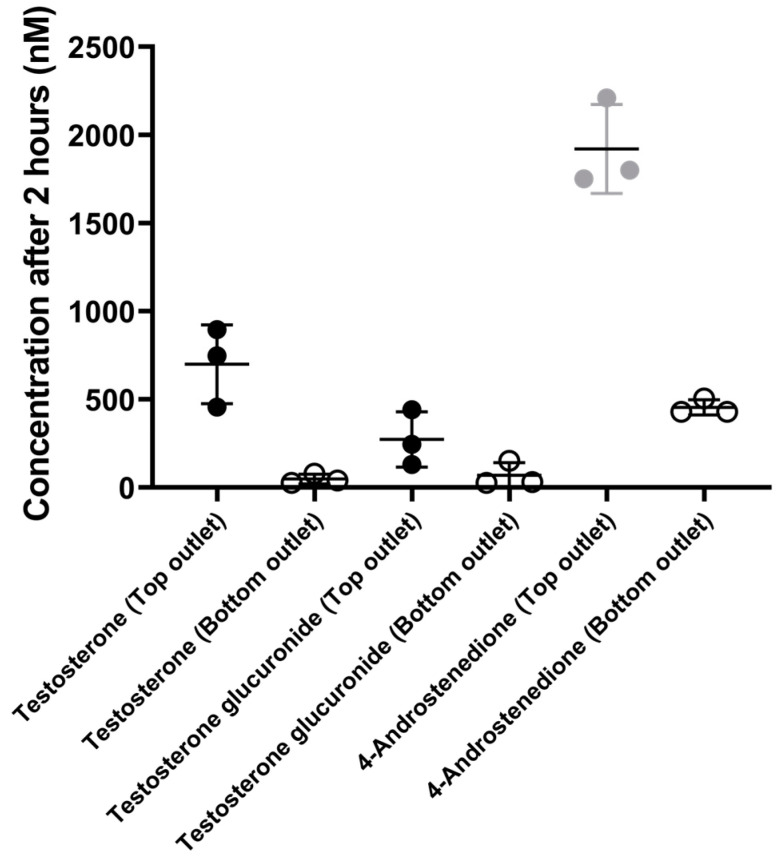
LC-MS/MS quantification of testosterone, 4-androstenedione, and testosterone glucuronide in the outlets of the Caco-2/HUVEC gut-on-chip 2 h after perfusing testosterone solution (10 µM). Data represent the mean ± SD from three biological replicates.

**Table 1 pharmaceutics-16-00296-t001:** PDMS ad- and absorptions of drugs with different physicochemical properties in cell-free organ-chips. cLogP and TPSA were calculated using the software MoKa version 2.6.4 (Molecular Discovery, Borehamwood, Hertfordshire, UK). Drugs were perfused as cocktails of 4 (10 µM per drug) simultaneously through the top and bottom channels of cell-free organ-chips via a 10 min introduction step at a flow rate of 600 µL/h. Outlet concentrations were determined after one hour (flow rate: 200 µL/h) and compared to the inlet concentrations. Data represent the mean values of two independent experiments with cell-free organ-chips.

Drug	MW	cLogP	TPSA	Compound Loss Bottom Channel (%)	Compound Loss Top Channel (%)
Apafant	456.0	0.98	72.6	8.0	15.0
Atorvastatin	558.6	4.46	111.8	7.4	18.9
Bazedoxifene	470.6	7.40	57.9	26.0	56.4
Busiprone	385.5	2.19	69.6	30.5	33.1
Elacridar	563.7	4.20	92.9	19.5	34.0
Ezetemibe	409.4	3.96	60.8	1.3	0.0
Felodipine	384.3	5.30	64.6	84.0	79.0
Indinavir	613.8	3.68	118.0	0.0	0.0
Midazolam	325.8	3.42	30.2	67.8	72.1
Nifedipine	346.3	3.13	107.8	25.6	28.3
Oxybutynin	357.5	4.87	49.8	100.0	99.9
Quinidine	324.4	2.79	45.6	13.4	0.3
Raloxifene	473.6	6.86	70.0	12.8	22.6
Rosuvastatin	481.5	1.90	140.9	7.9	12.6
Saquinavir	670.9	4.73	166.8	21.9	48.0
Testosterone	288.4	3.22	37.3	35.1	37.6
Zosuquidar	527.6	4.96	48.8	99.0	97.0

## Data Availability

Data are contained within the article and Appendix A.

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
