# Peer review of "Addressing the ADME Challenges of Compound Loss in a PDMS-Based Gut-on-Chip Microphysiological System"

_pharmaceutics, 2024, doi:10.3390/pharmaceutics16030296_

Round 1

Reviewer 1 Report

Comments and Suggestions for Authors

The authors aimed to utilize PDMS-based MPS for ADME studies and explored evaluation methods considering the adsorption and absorption of drugs to PDMS. The results demonstrated that, despite some loss due to adsorption and absorption, metabolic clearance could still be measured. This study is highly commendable as it is an objective assessment of an important issue in MPS by pharmaceutical users. The paper will be of great help to many users interested in MPS. The reviewer recommends acceptance of the paper after addressing the following concerns with additional explanations or modifications:

1. 2.5 Testosterone compound loss profiles: How were solutions within the channel recovered? It's conceivable that the media in the inlets and outlets were collected from the pods' chambers, but the method of recovery from the channels remains unclear.

2. Fig.3: The reason for testosterone concentration stabilizing at 7 uM is not clear. Concentrations at the top inlet and bottom inlet should also be shown. Were the inlets definitely at 10 uM? For the cell-free, diffusion from the top to the bottom through the porous membrane should be considered. Adsorption and absorption phenomena on the bottom side of the channel should also be expected. Please add descriptions regarding these points.

3. In this paper, testosterone levels were expressed in terms of concentration. However, in the Emulate system, since solutions are constantly flowing through the channels, it might be more appropriate to discuss in terms of quantity rather than concentration. Please consider this aspect.

4. In Fig.3B, the data for the Top channel at 120 minutes shows an unnaturally high spike with a large error bar. What is the reason for this?

5. 3.4. Testosterone metabolism in the Caco-2/HUVEC gut-on-chip: Testosterone metabolites were measured, but was the mass balance with the parent compound accurate? Adsorption and absorption of the metabolites should also be considered. Please include descriptions about these aspects.

6. The validity of apparent clearance and papp calculated in section 3.5 should also be discussed. For example, a comparison with conventional culture insert-based experimental systems should be considered.

7. For practical use, predicting substances with varying rates of adsorption and absorption is challenging. The cell-free data in Fig.6C suggests that treating with a testosterone solution for 2 hours could saturate the adsorption and absorption phenomena to PDMS. This point should be included in the discussion.

8. Discussion: The authors state, "Since most of the drugs didn’t show this discrepancy, difference in channel geometry was most likely not the reason." However, channel geometry is a major factor affecting adsorption and absorption phenomena. Emulate's chips have different flow channel sizes for top and bottom, resulting in different surface area ratios per unit volume. With varying channel cross-sectional areas, the flow rates differ even if the flow volume remains constant. Could the varying loss amounts for different drugs be due to differences in adsorption and absorption speeds to PDMS? Please consider this from the perspectives of surface area ratio and flow rate.

Reviewer 2 Report

Comments and Suggestions for Authors

Patrick Carius et al presented a method to evaluate Papp in PDMS-based Gut-on-chip microphysiological system while acknowledging the compound loss due to drug ad- and absorption to PDMS. The manuscript is well structured with experimental results and discussion on limitations of the method. Please address the following concerns and make corresponding edits.

1.               Figure 2: Please elaborate on the difference in Papp of Apafant under different conditions. If the efflux of Apafant in the Caco-2/HUVEC gut-on-chip is 2-fold higher than that in Caco-2 on Transwell inserts, then the P-gp activities on two devices are quite different, instead of “similar”. What ratios are usually considered “acceptable” when evaluating the barrier integrity? The authors should provide some references (i.e. results from other gut-on-chips) to support the claim.

2.               Why are different flow rates are used in Figure 3 (600 μL/h) and Figure 5 (200 μL/h)? What’s the considerations here?

3.               Figure 6: Please make it clear that 6A and 6B are results from Cell-free organ-chips and Caco-2/HUVEC gut-on-chip, respectively, in the figure and figure captions.

Round 2

Reviewer 1 Report

Comments and Suggestions for Authors

The authors have addressed all the reviewers' concerns and the manuscript is now suitable for publication in the journal.

著者らは査読者の懸念をすべて解決し、原稿はジャーナルに掲載するのに適したものになりました。